# Evaluation of Recurrence Risk in Irreversible Electroporation-Treated Pancreatic Adenocarcinoma Patients Using Radiomics Signatures

**DOI:** 10.3390/cancers17142338

**Published:** 2025-07-15

**Authors:** Jacob W. H. Gordon, Akshay Goel, Robert C. G. Martin

**Affiliations:** 1Tempus AI, Chicago, IL 60654, USA; jacob.gordon@tempus.com (J.W.H.G.); akshay.goel@tempus.com (A.G.); 2Division of Surgical Oncology, Department of Surgery, University of Louisville, Louisville, KY 40202, USA

**Keywords:** radiomics, pancreatic adenocarcinoma, irreversible electroporation, machine learning, recurrence

## Abstract

To investigate if radiomics signatures generated from longitudinal CT scans could predict IRE treatment effectiveness and outcomes in patients with locally advanced pancreatic cancer (LAPC).  A cohort of 50 (60% male, mean [SD] age 60.7 [8.7] years) LAPC patients treated with IRE were retrospectively selected. Preoperative and 12-week follow-up CT were reviewed by two radiologists for tumor segmentation. Statistically significant separation between high and low patient TTR risk groups was observed in: Gray-level co-occurrence matrix (HR = 2.65, *p* < 0.01, median survival difference = 6.6 mo); composite radiomics features derived from the following feature groups: all radiomics features (HR = 2.27, *p* = 0.01, median survival difference = 6.4 mo), intensity features (HR = 3.13, *p* < 0.01, median survival difference = 14.0 mo), and filter features (HR = 2.27, *p* = 0.01, median survival difference = 6.4 mo).  Pre-treatment radiomics signatures were significantly associated with LAPC patient outcomes. The observed correlations used pre-treatment CT scans, implying that the features are predicting individual risk of disease recurrence.

## 1. Introduction

Pancreatic ductal adenocarcinoma (PDAC) is the third most deadly type of cancer within the United States, while it is only the eleventh most common. An estimated 56,700 new individuals are diagnosed each year, along with 45,700 new deaths [1,2]. Surgical resection resulting in negative tumor margins combined with additional multidisciplinary treatment regimens currently provides the best treatment outcomes [3].

However, locally advanced pancreatic cancer (LAPC) has historically been deemed unresectable and thought of as a continuum of metastatic disease. Fortunately, additional consolidative treatment options for LAPC following induction chemotherapy exist and have been successfully utilized with improved outcomes. Irreversible electroporation (IRE), a non-thermal ablation technology, has begun to gain acceptance within the last decade [4,5,6]. IRE induces cellular apoptosis without disrupting surrounding tissue structural integrity [7]. Martin et al. demonstrated that IRE is a safe and effective treatment of LAPC with initial improvements in median overall survival (OS) to 25.3 months [8]. These results were confirmed with a combination of chemotherapy and IRE, improving the median OS to 30.7 months, critically implicating IRE to be included in the multimodal treatment of LAPC [9].

Pancreatic tumors are even more challenging to evaluate because of the fibrotic stroma desmoplasia, characterized by the extensive deposition of extracellular matrix (ECM) components and the localization and activation of cancer-associated fibroblasts, which reduces vasculature patency and drug access and alters the anti-tumor immune response. Therefore, assessing the tumor response using RECIST 1.1 lacks specificity and accuracy. In addition, the interval response after certain recent (<4 months) oncologic therapies of the targeted area can demonstrate an increase in the size of the tumor despite the initial perceived complete destruction. Therefore, size-specific criteria alone do not adequately represent the success of certain oncologic treatments in pancreatic cancer. This leaves assessing the efficacy of LAPC an unmet need, which could potentially be aided by radiomics.

Radiomics is a method of transforming regions of interest (ROI) within radiology scans into numerical features which describe the ROI’s shape, voxel densities, and voxel textures objectively and quantitatively. Recent studies have shown that radiomics features correlate with tumor genotypes and can predict patient outcomes within several different disease sites, including the lung, liver, and pancreas [10,11]. While other metrics of quantifying imaging such as the delta and interface score have been shown to reliably predict outcomes [12], these techniques are less objective than radiomics and represent a less complete featurized description of the image and the ROI.

Delta radiomics is an advanced method that utilizes the changes in an ROI across longitudinal exams to illustrate differences in radiomics features over time. It has been shown to correlate with the early evaluation of the response to pancreatectomy after chemoradiotherapy in pancreatic cancer patients [13].

We hypothesize that by using delta radiomics, which leverages the longitudinal information captured from radiomics features in both pre-treatment and follow-up scans, we can identify patients who respond effectively to IRE therapy. Furthermore, the application of radiomics techniques to LAPC patients could offer a non-invasive, pre-surgical method to evaluate candidates for IRE treatment, utilizing only pre-treatment scans. This study aims to evaluate both radiomics and delta radiomics as possible predictors of recurrence outcomes in patients treated with an IRE by separating patients into high- and low-risk (of going on to experience a recurrence event) groups.

## 2. Materials and Methods

### 2.1. Data Collection and Generation

#### 2.1.1. Cohort Selection

The cohort includes data from an Institutional Review Board (IRB)-approved, single-arm study of patients diagnosed with borderline resectable PDAC (BRPC) according to the National Comprehensive Cancer Network (NCCN) guidelines between July 2015 and September 2020 at [BLINDED for review purposes]. Patients treated by pancreatic resection and IRE were compared to the same-stage patients treated with pancreatic resection alone. This is a retrospective analysis of our prospective pancreatic cancer database of patients treated with and without IRE for margin accentuation for BRPC ([13] [BLINDED]). All patients provided written informed consent. A diagnosis of BRPC disease was established by biopsy-proven adenocarcinoma of the pancreas with reconstructable venous involvement or less than 180° encasement of their superior mesenteric artery (SMA) or celiac artery without evidence of metastatic lesions [9,14,15]. Staging included triple-phase CT scans with less than 1.5 mm cuts at the time of diagnosis and repeated 1–2 weeks prior to resection [8,16].

Inclusion criteria for IRE therapy included patients who underwent induction therapy consisting of chemotherapy and/or external beam radiation therapy, were diagnosed with non-metastatic LAPC, and were without disease progression 4–6 months after induction therapy. Eligible patients received either IRE with resection or resection alone based on the surgeon’s discretion and/or intra-operative findings at the time of dissection. Patients underwent restaging evaluation 1 to 2 weeks after induction therapy via triple-phase CT scan and serum tumor markers. Patients with implanted cardiac pacemakers or defibrillators unable to be deactivated, non-removable implants with metal parts within 1 cm of the target lesion, a myocardial infarction within 3 months, or unsuitable for general endotracheal anesthesia were excluded. For this analysis, a cohort of 50 consecutive stage II patients with BRPC based on preoperative imaging was selected.

#### 2.1.2. Data Annotation

The segmentation of tumors and normal pancreatic parenchyma was performed by four radiology fellows with a minimum of seven years of clinical experience and reviewed by a board-certified diagnostic body radiologist using the open-source software package ITK-SNAP Version 4.2.2 [17]. The entire lesion, including any extra-pancreatic extensions, was annotated. Vessels and stents were subtracted from the final contours. In post-surgical CT scans, post-IRE-treatment changes were also annotated. Many patients were missing either the arterial or portal venous phase contrast scans from their triple-phase acquisitions. In total, 44% of the annotated scans were arterial phase, while the other 56% were portal venous. Examples of pre-surgical and 12-week follow-up annotations are shown in Figure 1.

#### 2.1.3. Feature Generation

A total of 2078 radiomics features were extracted from the annotated tumor ROI in pre-treatment and 12-week follow-up CT scans. Feature extraction was performed using the [BLINDED] ([BLINDED]) feature extraction software. These features can be categorized into 5 groups: shape (16), intensity (18), texture (68), local texture (86), and filter (1892) features. Shape features describe the shape and size of the ROI. Intensity features were derived from statistics of the ROI’s voxels densities. Texture features aim to capture subtle changes and patterns in the ROI’s densities. The types of texture features extracted for this experiment were the gray-level co-occurrence matrix (GLCM), the gray-level run length matrix (GLRLM), the gray-level size zone matrix (GLSZM), and the gray-level dependence matrix (GLDM). Local texture features focus specifically on local changes in the ROI’s voxel densities. These consist of local binary pattern (LBP) and local ternary pattern (LTP) features, as described in Gore S. et al. [18]. Filter features were extracted from CT images after Log Sigma, Wavelet, Wavelet 2, Logarithm, Exponential, and Square filters were applied.

### 2.2. Clinical Endpoint

To evaluate the effects of IRE therapy on patients, we examined several survival- and recurrence-based endpoints. Recurrence was defined as per RECIST 1.1, either a 20% increase on the diameter of the IRE-treated pancreatic tumor or any new extra-pancreatic lesion/tumor that was greater than 1.0 cm in size. The analysis primarily focused on time to recurrence (TTR) but also applied the same methodologies to time to local recurrence (TTLR), time to distant recurrence (TTDR), overall survival (OS), and recurrence-free survival (RFS). The first step of the analysis binarized these endpoints across a time threshold, simplifying the analysis and allowing us to establish groups of patients at high and low risk of an event (discussed further in Section 2.4). Time thresholds were selected to create as even a class split as possible from 6-month, 12-month, and 18-month options. A 12-month time threshold was used for TTR and RFS. An 18-month time threshold was used for TTDR, TTLR, and OS. Patients whose date of last follow-up was before the time threshold, and had yet to experience a relevant event, were censored from our analysis. The class splits between different endpoints and the number of patients censored at 6-month, 12-month, and 18-month time thresholds are shown in Table 1 for the 49 patients with viable pre-treatment CT scans.

Table 1 shows the patients sorted into each class for all endpoints and the evaluated time thresholds. Only a single time threshold was analyzed for each endpoint. The grayed-out rows are the time thresholds that were not analyzed. Patients who experienced an event—any type of recurrence—before the given time threshold were sorted into the “event-recurrence class”. Patients who did not experience an event before the given time threshold but were followed until the time threshold were sorted into the “no event class”. Patients who were not followed up to the time threshold were censored from any further analysis. Overall survival = OS; recurrence-free survival = RFS; time to recurrence = TTR; time to local recurrence = TTLR; time to distant recurrence = TTDR.

### 2.3. Radiomics Analysis

Tests were performed using two sets of radiomics features. First, a small hand-selected set of radiomics features (Appendix A). Second, principal component analysis (PCA) was applied to groups of features to create a smaller set of representative features, limiting the number of tests while leveraging all 2078 generated radiomics features. We refer to these PCA-generated linear combinations as composite radiomics (CR) features (Appendix A) [19].

Both of these feature sets were then correlated with binary survival endpoints to separate patients into high- and low-risk (of event) groups. Then, a survival analysis was performed to confirm and quantify any observed correlations with binary survival endpoints.

Each of these feature sets is discussed in more detail in Appendix A. Tests were performed using radiomics features extracted from both the pre-IRE and post-IRE CT scans separately, as well as from the delta radiomics features generated by looking at the difference between the post-IRE and pre-IRE radiomics features.

### 2.4. Statistics

All statistical analysis was performed in Python Version 3.13.0 using the SciPy (v1.7.1) [20], Lifelines (v0.26.0) [21], and SciKit-Learn (v0.24.2) [22] Python packages. PCA was used for dimensionality reduction.

All radiomics analysis results were evaluated using Mann–Whitney U Rank Sum tests along with the area under the receiver operating characteristic curve (AUC) to evaluate if a feature could successfully separate binary patient outcomes into high- and low-risk groups. The Mann–Whitney U Rank Sum test was used instead of a standard *t*-test because the probability distribution functions of radiomics features are often not normally distributed. *p*-values below 0.05 were considered statistically significant.

Radiomics analysis results were then further analyzed using survival analysis and Kaplan–Meier (KM) curves. For the selected radiomics and CR features, patients were sorted into high-risk and low-risk groups by maximizing the sensitivity and specificity of the features’ “prediction” of the binary endpoint. Risk groups were evaluated to determine if they represented patients with statistically significant and meaningfully separated patient outcomes. The significance was evaluated using a log-rank test, and the strength of the separation was evaluated by calculating the hazard ratio (HR).

A correlation was only considered significant if both the Mann–Whitney U *p*-value and the log-rank test *p*-value fell below the threshold of significance, 0.05.

Our approach utilized multi-step feature selection pipeline univariate filtering, followed by regularized regression to mitigate overfitting and reduce the risk of spurious associations. These steps were designed to prioritize model generalizability rather than rely on individual feature significance. Moreover, our models were validated using bootstrap, which provides a more robust safeguard against Type I error than the correction of *p*-values alone.

## 3. Results

### 3.1. Cohort Demographics

The cohort of 50 patients was 60% male, 96% Caucasian, 2% African American, and 2% Pacific Islander. The mean [SD] age of the cohort was 60.7 [8.7] years, with a range of ages of 38–80 years (men: mean [SD] = 62.3 [7.9], range = 38–80; women: mean [SD] = 58.4 [9.3], range = 38–73). A total of 35 patients received IRE therapy alone, while the other 15 received IRE therapy along with a pancreatectomy.

One patient’s baseline scan and a different patient’s 12-week follow-up scans were of poor quality and needed to be excluded from the radiomics analysis. This resulted in a cohort of 49 patients with pre-treatment radiomics features, 49 patients with 12-week follow-up radiomics features, and 48 patients with delta radiomics features.

All patients were followed for at least 18 months or until their death, and the mean [SD] time to the last follow-up was 23.2 [13.7] months. In the full cohort, 25 (50%) recurred within 12 months, while 8 had died.

### 3.2. Select Radiomics Feature Results

Of our select radiomics features, only GLCM sum entropy showed statistically significant correlations with any of the tested endpoints or feature sets. Table 2 shows several key statistics for significantly correlated features. A full table of all results from select radiomics features can be found in Appendix A.

Correlations were only observed when using features extracted from pre-treatment CT scans and not from follow-up CT scans or the delta radiomics features created with the difference between the two scans. We observed the same significance for the TTR and RFS endpoints. No other endpoints produced any significant results.

The most significant observed correlation is between GLCM sum entropy extracted from the pre-treatment tumor burden and RFS (Mann–Whitney U *p*-value < 0.005, AUC = 0.74, HR = 2.68, log-rank *p*-value < 0.005, median survival difference = 6.41 m).

Table 2 contains significant results from the examined selected radiomics features. The endpoint being compared against is identified in the Endpoint column. The image source used to generate the radiomics features is identified by the Feature Type column (pre-treatment scan, 12-week follow-up scan, or delta radiomics features generated from both). The radiomics feature being compared is identified by the Feature column. For each of these comparisons, the feature’s mean and standard deviation (STD) are given. The AUC from an ROC plot, the associated optimal threshold, and the Mann–Whitney U *p*-value are given in the following three columns. The remaining four columns, the log-rank *p*-value, the hazard ratio, the concordance index, and the median survival difference (in months), are generated from a survival analysis of the high- and low-risk groups created by applying the optimal threshold. Notably, 12-month binarized time to recurrence = 12 m TTR; 12-month binarized recurrence-free survival = 12 m RFS; pre-treatment scan radiomics features = pre; gray-level co-occurrence matrix = GLCM.

### 3.3. Composite Radiomics Feature Results

In general, more CR features were correlated with survival outcomes than selected radiomics features. Correlations between TTR and RFS were observed from both the CR features examined by the larger full feature group and filtered feature group, as well as with intensity CR feature 1. Texture CR feature 1 was also only barely below the significance criteria for both TTR and RFS. Texture and intensity features appear in both their own feature groups and the larger filtered and full feature groups; therefore, texture and intensity features appear to be the driving force behind the observed correlations. All correlations to TTR and RFS used features generated from the pre-treatment CT scans. No significant correlations to either TTR or RFS were observed using the radiomics features generated from 12-week follow-up scans or delta radiomics features.

No statistically significant correlations with TTLR or OS were observed. Statistically significant correlations with TTDR were observed, but they were less consistent than the correlations to TTR and RFS. The full and filtered feature groups show correlations with TTDR using the pre-treatment CT scans, similar to the observed correlations with TTR and RFS discussed in the previous paragraph, but only from their second CR features, not both.

Shape CR feature 1, extracted from the 12-week follow-up scan, significantly correlates with TTDR. This is the only correlation observed that uses radiomics features extracted from the 12-week follow-up scan, and the only observed correlation using the shape radiomics features.

We also see two correlations to TTDR using delta radiomics features, from texture CR feature 2 and filter CR feature 2.

Local texture CR features produced no significant correlation with any endpoints.

All significant results can be found in Table 3. For a complete table of results, see Appendix A.

Table 3 contains the significant results from the examined composite radiomics (CR) features. The endpoint being compared against is identified in the Endpoint column. The image source used to generate the radiomics features is identified by the Feature Type column (pre-treatment scan, 12-week follow-up scan, or delta radiomics features generated from both). The CR feature being compared is identified by the Feature column. For each of these comparisons, the feature’s mean and standard deviation (STD) are given. The AUC from an ROC plot, the associated optimal threshold, and the associated Mann–Whitney U *p*-value are given in the following three columns. The remaining four columns, the log-rank *p*-value, the hazard ratio, the concordance index, and the median survival difference (in months), are generated from a survival analysis of the high- and low-risk groups created by applying the optimal threshold. Notably, 12-month binarized time to recurrence = 12 m TTR; 18-month binarized time to distant recurrence = 18 m TTDR; 12-month binarized recurrence-free survival = 12 m RFS, pre-treatment scan radiomics features = pre; 12-week follow-up scan radiomics features = 12 w; delta radiomics features = delta.

The median overall survival in the IRE-treated group was 33.3 months from diagnosis, with a relapse-free survival of 16.8 months. In univariate Kaplan–Meier (KM) analyses, patients who underwent pancreatectomy with IRE were compared to patients who underwent IRE alone; there was a similar median overall progression-free survival from diagnosis with resection with an IRE of 21.4 (11.1 to 24.3) versus an IRE only of 22.6 (19.9 two 25.4) months, *p* = 0.0690. Overall survival from diagnosis for pancreatectomy with IRE was 24.9 (12.6 to 39.1) versus an IRE only of 29.4 (23.1 to 36.2) months, *p* = 0.23.

## 4. Discussion

The results of this study show that some intensity and texture features can individually correlate with patient outcomes after IRE therapy, but also that there is additive information available from combining those feature sets and applying various filters to the original CT image to draw out new information.

These results have only been tested on IRE-treated patients but may apply more generally to PDAC patients and correlate more to the aggressiveness of the specific patient’s cancer than their response to a specific treatment. Similar data have been published on the use of these features for predicting recurrences and overall survival [23,24]. This is further reinforced by the lack of predictive correlations observed related to local recurrences. If the observed radiomics correlations were related to the efficacy of a surgical procedure, IRE or otherwise, we would expect those correlations to extend from TTR to TTLR, which they do not. These predictive results need to be further evaluated on larger cohorts of patients and tested for generalizability to other treatment regimens.

In general, we observed the PCA-generated CR feature-based methods, which utilize more input radiomics features, while not individually producing more significant or stronger correlations, more reliably produce correlations with RFS and TTR than the hand-selected radiomics features we used. This approach is not only useful in identifying contributing component radiomics features without introducing any human bias when selecting features to examine but also may leverage orthogonal predictive power from many distinct radiomics features within our full set of 2078 radiomics features. Future work can dig into the CR features’ component parts to identify other individual radiomics features of interest or build custom composite features to better correlate with patient outcomes. Additionally, because the shape and local texture features appear to have minimal signal, excluding those feature groups could produce a more refined PCA-generated CR feature. Having an additional independent test cohort would also let us test individual features for generalizability and use a feature’s robustness against imaging or annotation variations as an inclusion criterion. For example, patients with their primary tumors in place, who have undergone radiation therapy, could be evaluated to refine these features. This would help further ensure generalizability to new distinct cohorts in the future.

Despite our hypothesis, delta radiomics features and follow-up CT scans added little to no benefit in identifying patients with worse outcomes. It is possible that due to the aggressive nature of IRE therapy, the changes to the treated region destroy and disrupt any previously present radiomics correlations to a patient’s outcome. Irreversible electroporation (IRE) induces local tissue changes such as edema, necrosis, and gas formation, which can create artifacts and alter tissue contrast on post-procedural imaging. These artifacts may reduce the consistency and reproducibility of radiomics features by introducing noise or disrupting standard tissue patterns. Additionally, anatomical distortion post-IRE—such as shifts in organ boundaries, the collapse or expansion of treated areas, and deformation due to inflammation—can compromise spatial alignment and segmentation accuracy. This may significantly affect delta radiomics, which rely on precise voxel-wise comparison between pre- and post-treatment images. It is possible that the temporal window of post-treatment imaging may not be optimal for capturing meaningful biological changes, potentially contributing to the diminished discriminative power of delta features. It is also possible that, given our limited sample size, we are underpowered to leverage additional information in the more complex and noisier delta radiomics features.

This study is limited in several ways. The initial cohort is only 50 patients, and some of them were treated with additional surgical procedures other than an IRE, possibly contaminating or complicating our results. The limited cohort size prevents the application of more sophisticated supervised machine learning approaches. Validation of the observed correlations on an independent, unseen cohort of patients is required to test for generalizability. Additionally, to properly test whether the radiomics signatures we identified predict IRE treatment response or the patient’s more general prognosis, a second control cohort of patients not treated with an IRE or preferably any other surgical procedures would be required. However, considering the breadth of significant correlations we observed between radiomics and RFS/TTR, there is great potential for the future development of more sophisticated models using a larger cohort of patients.

## 5. Conclusions

Overall, we found that radiomics features, excluding shape and local texture features, show promise in evaluating patient prognosis with an entirely non-invasive pre-treatment method. We observe no evidence that these correlations are specific to IRE-treated patients. While larger, more varied evaluation cohorts are required to evaluate the accuracy, reliability, and generalizability of these techniques, these early results are promising.

## Figures and Tables

**Figure 1 cancers-17-02338-f001:**
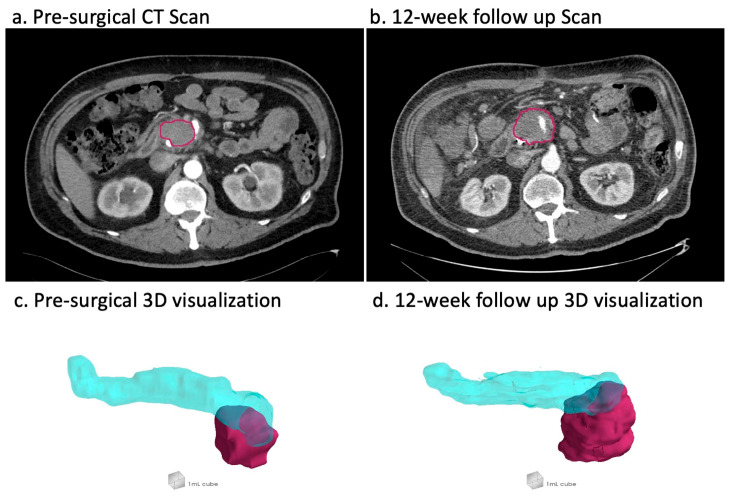
Shows examples of pre-surgical and post-surgical tumor/treated region annotations as well as 3D visualizations of the annotated tumor and pancreatic tissue in axial slices of arterial contrast phase scans from a 64-year-old male patient. The tumor/post-treatment changes are annotated in red in both the slices and the 3D visualization, while the rest of the pancreas is annotated in blue in the 3D visualization.

**Table 1 cancers-17-02338-t001:** Endpoints and class splits.

Endpoint	Time Threshold	Event Class	No Event Class	Censored
OS	6 months	1	48	0
OS	12 months	8	41	0
OS	18 months	20	29	0
RFS	6 months	5	44	0
RFS	12 months	28	21	0
RFS	18 months	36	13	0
TTR	6 months	5	44	0
TTR	12 months	25	21	3
TTR	18 months	32	13	4
TTLR	6 months	3	45	1
TTLR	12 months	14	29	6
TTLR	18 months	18	19	12
TTDR	6 months	3	46	0
TTDR	12 months	14	30	5
TTDR	18 months	22	16	11

**Table 2 cancers-17-02338-t002:** Selected radiomics features’ significant results.

Endpoint	Feature Type	Feature	Mean	STD	Mann–Whitney U *p*-Value	AUC	Optimal Threshold	Log-Rank *p*-Value	Hazard Ratio	Concordance Index	Median Survival Difference [Months]
12 m TTR	pre	GLCM Sum Entropy	3.11	0.44	0.005	0.26	3.23	0.005	2.65	0.65	6.61
12 m RFS	pre	GLCM Sum Entropy	3.10	0.44	0.005	0.26	3.23	0.005	2.68	0.65	6.41

**Table 3 cancers-17-02338-t003:** Composite radiomics features’ significant results.

Endpoint	Feature Source	Feature	Mean	STD	Mann–Whitney U *p*-Value	AUC	Optimal Threshold	Log-Rank *p*-Value	Hazard Ratio	Concordance Index	Median Survival Difference [Months]
12 m TTR	pre	Full CR Feature 1	0.00	27.51	0.014	0.29	−7.50	0.024	2.10	0.61	5.33
		Full CR Feature 2	0.00	16.32	0.002	0.76	−0.38	0.012	2.27	0.62	6.41
		Intensity CR Feature 1	0.00	3.13	0.034	0.32	1.54	0.008	3.13	0.62	14.01
		Filter CR Feature 1	0.00	26.83	0.014	0.29	−8.08	0.038	1.98	0.60	4.44
		Filter CR Feature 2	0.00	15.80	0.002	0.77	−0.64	0.012	2.27	0.62	6.41
18 m TTDR	pre	Full CR Feature 2	0.00	16.79	0.040	0.70	2.90	0.002	3.02	0.65	6.41
		Filter CR Feature 2	0.00	16.27	0.032	0.71	−1.94	0.010	2.51	0.64	9.63
	12 w	Shape CR Feature 1	0.00	3.09	0.024	0.72	0.25	0.005	2.76	0.64	6.58
	delta	Texture CR Feature 1	0.00	3.80	0.048	0.31	−0.43	0.008	2.66	0.63	10.78
		Filter CR Feature 2	0.00	15.66	0.041	0.70	9.85	0.000	5.02	0.63	5.98
12 m RFS	pre	Full CR Feature 1	0.00	27.08	0.014	0.29	2.56	0.033	2.17	0.61	4.60
		Full CR Feature 2	0.00	16.57	0.012	0.71	5.25	0.004	2.68	0.63	5.72
		Intensity CR Feature 1	0.00	3.07	0.040	0.33	1.08	0.011	3.01	0.61	13.78
		Filter CR Feature 1	0.00	26.43	0.015	0.29	2.51	0.033	2.17	0.61	4.60
		Filter CR Feature 2	0.00	15.94	0.013	0.71	5.23	0.004	2.68	0.63	5.72

## Data Availability

All data is available on request to the corresponding author.

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
