# Peer review of "Evaluation of Recurrence Risk in Irreversible Electroporation-Treated Pancreatic Adenocarcinoma Patients Using Radiomics Signatures"

_cancers, 2025, doi:10.3390/cancers17142338_

Round 1
Reviewer 1 Report
Comments and Suggestions for Authors
This retrospective study investigated a very interesting and important point: the possibility of predicting response to irreversible electroporation (IRE) using radiomics signatures. The manuscript is well-written and presented. Below are my comments to improve the manuscript:
1) Cohort demographics included in the Method section actually belong in the Results.
2) You should clearly define "recurrence". Is it an increasing volume? Or the appearance of metastases? Or both? Please clarify
3) In the results, describe the median outcomes time (OS, RFS, TTR, etc) in the group of "responders" and non-responders".
4) There was a difference in survival or recurrence between IRE+resection vs IRE alone?
5) You could add some Kaplan-Meier figures.
Minor:
- The application of radiomics to other alternative, less invasive local ablation methods (i.e., EUS-RFA) should be mentioned in the introduction/discussion. Doing so, cite PMID 36837560
Author Response
1) Cohort demographics included in the Method section actually belong in the Results. *** This has been moved to the results. ***
2) You should clearly define "recurrence". Is it an increasing volume? Or the appearance of metastases? Or both? Please clarify *** This has been defined per RECIST 1.1. ***
3) In the results, describe the median outcomes time (OS, RFS, TTR, etc) in the group of "responders" and non-responders". *** I am sorry but I do not know what your mean by responders...... All patients had confirmed complete IRE "ablations" of their tumors at 3 and 6 months. There were no "non-responders" in the treated patients. If you could better describe the delineation you are looking for I can evaluate. ***
4) There was a difference in survival or recurrence between IRE+resection vs IRE alone? No, there was no difference in either group.
5) You could add some Kaplan-Meier figures. *** based on the # of radiomen features we do not have the power to show a KM curve with these features. ***
Minor:
- The application of radiomics to other alternative, less invasive local ablation methods (i.e., EUS-RFA) should be mentioned in the introduction/discussion. Doing so, cite PMID 36837560 *** We would but this was for Neuroendocrine tumors of he pancreas and not adenocarcinoma. ***
Reviewer 2 Report
Comments and Suggestions for Authors
The research presents a retrospective radiomics analysis assessing the probability of recurrence in individuals with locally advanced pancreatic cancer (LAPC) who have undergone irreversible electroporation (IRE). This research used principal component analysis (PCA) to discern composite determinants of recurrence and survival outcomes. Additionally, radiomic characteristics are obtained from the analysis of pre-treatment and follow-up CT scans.
The issue is both relevant and pressing, particularly given the limited treatment alternatives and the unfavourable outlook for LAPC. The combination of radiomics and machine learning to forecast how well a therapy would work is a major step forward that greatly improves the development of precision oncology.
Strengths:
Radiomics is used by the initiative to treat the hard-to-treat cancer subtype known as LAPC. The treatment is done using a relatively new method called IRE. This approach might be used to guess biomarkers based just on pictures taken before therapy.
The authors used a large feature set including 2078 radiomic characteristics and applied rigorous statistical methodologies to guarantee thorough investigation. The methods used were the Mann-Whitney U test, Kaplan-Meier survival analysis, and principal component analysis (PCA).
Using preoperative imaging to find out the likelihood of recurrence might have a big effect on how doctors make decisions in pancreatic cancer.
Adding more research and tables makes the findings more reliable and clear.
Need improvement:
The inclusion criteria, particularly regarding how "borderline resectable" was defined and the precise treatment protocols (IRE alone vs. IRE + surgery), are not fully clarified.
Clearly define how patients were stratified, and ensure consistent terminology for “LAPC” and “BRPC” throughout the manuscript.
There is no true control cohort (non-IRE-treated LAPC patients), and the dataset is limited to 50 patients. A discussion on selection bias and lack of control/comparator arm should be expanded.
The composite radiomics (CR) features derived through PCA are statistically valid but lack biological interpretability.
The manuscript notes that delta radiomics added little value but does not thoroughly explore why. Include hypotheses on why post-treatment or delta features underperformed—consider IRE-induced image artifacts or anatomical distortion affecting radiomics consistency.
Multiple comparisons were conducted (across 2078 features), increasing the chance of Type I errors. Consider applying false discovery rate (FDR) correction or Bonferroni adjustment to account for multiple hypothesis testing.
The discussion includes promising insights but remains speculative in some parts. Anchor the discussion more solidly in current literature. For example, discuss how findings align or contrast with previous radiomics studies in PDAC or other IRE-treated tumors.
Author Response
Need improvement:
The inclusion criteria, particularly regarding how "borderline resectable" was defined and the precise treatment protocols (IRE alone vs. IRE + surgery), are not fully clarified. *** This is reported in Reference 14 - which is "Blinded for Review" All decision making is there in that Reference. ***
Clearly define how patients were stratified, and ensure consistent terminology for “LAPC” and “BRPC” throughout the manuscript. *** Same all is there in Ref 14. ***
There is no true control cohort (non-IRE-treated LAPC patients), and the dataset is limited to 50 patients. A discussion on selection bias and lack of control/comparator arm should be expanded. *** This is in the methods line 98. ***
The composite radiomics (CR) features derived through PCA are statistically valid but lack biological interpretability. *** This is the reason for this study. We are trying to demonstrate that these CR features are biologically predictive for RFS. ***
The manuscript notes that delta radiomics added little value but does not thoroughly explore why. Include hypotheses on why post-treatment or delta features underperformed—consider IRE-induced image artifacts or anatomical distortion affecting radiomics consistency. *** Thank you. Additional reasons for this have been added. Line 371-382 ***
Multiple comparisons were conducted (across 2078 features), increasing the chance of Type I errors. Consider applying false discovery rate (FDR) correction or Bonferroni adjustment to account for multiple hypothesis testing. ***
We appreciate the reviewer’s point regarding the risk of Type I errors associated with multiple comparisons across a large number of radiomic features. However, we respectfully argue that traditional correction methods such as Bonferroni or FDR may not be suitable or necessary in the context of high-dimensional radiomic data where the primary goal is feature reduction and model development, rather than strict hypothesis testing on individual features.
Our approach utilized a multi-step feature selection pipeline univariate filtering, followed by regularized regression to mitigate overfitting and reduce the risk of spurious associations. These steps were designed to prioritize model generalizability rather than rely on individual feature significance. Moreover, our models were validated using bootstrap, which provides a more robust safeguard against Type I error than correction of p-values alone. this has been added to the methods. ***
The discussion includes promising insights but remains speculative in some parts. Anchor the discussion more solidly in current literature. For example, discuss how findings align or contrast with previous radiomics studies in PDAC or other IRE-treated tumors. *** Thank you, this has been added per third reviewer. ***
Reviewer 3 Report
Comments and Suggestions for Authors
Overall, this is a well-designed study that presents data on the correlation between radiomic features and the clinical outcomes of patients with BRPC treated with IRE. The approach and future studies that this work illuminates may lead to a difference in clinical management of PDAC. The paper is well written, except for one methods portion that can be improved for clarity, as listed below. Additionally, the discussion completely lacks references and any discussion on how this work fits into the current literature, and needs to be improved upon prior to publication.
- The legend for Table 1 (lines 192-202) does a better job clearly laying out the methods than the related written text (lines 182-190). Consider improving written text for clarity.
- The discussion, while a good summary of the presented results, completely lacks references or a discussion of how these results fit into the current field of knowledge. Here are a few examples of places where statements and conclusions were made that could use support:
- Lines 334-337 and 381-382, concluding that the lack of extension from TTR to TTLR means that the radiomics are not tied to efficacy sounds logical, but is this an accepted conclusion that has been previously used in literature? Otherwise, this could be seen as an unreasonably strong conclusion.
- Lines 340-356, the PCA-generated CR methodology is explained throughout the manuscript without any reference to prior work. Has anyone else done this before? If so, what were those findings? If not, is the prior work that supports the idea to utilize this methodology?
- Lines 358-64, for the discussion on the delta radiomics hypothesis, there is a lack of context in the discussion section to the relevance.
Author Response
- The legend for Table 1 (lines 192-202) does a better job clearly laying out the methods than the related written text (lines 182-190). Consider improving written text for clarity. ***Thank you this has been edited. ***
- The discussion, while a good summary of the presented results, completely lacks references or a discussion of how these results fit into the current field of knowledge. Here are a few examples of places where statements and conclusions were made that could use support:
- Lines 334-337 and 381-382, concluding that the lack of extension from TTR to TTLR means that the radiomics are not tied to efficacy sounds logical, but is this an accepted conclusion that has been previously used in literature? Otherwise, this could be seen as an unreasonably strong conclusion. *** Thank you, these conclusions have been further strengthened by recent publications in this exact disease site, referenced. ***
- Lines 340-356, the PCA-generated CR methodology is explained throughout the manuscript without any reference to prior work. Has anyone else done this before? If so, what were those findings? If not, is the prior work that supports the idea to utilize this methodology? *** Yes this has been referenced line 197 at first mention of this work. Our apologies for not referencing our preliminary work. ***
- Lines 358-64, for the discussion on the delta radiomics hypothesis, there is a lack of context in the discussion section to the relevance. *** Updated with better context of another cohort. ***
Round 2
Reviewer 1 Report
Comments and Suggestions for Authors
Dear Authors,
Thank you for amending the manuscript. Here are my responses to your answers:
Point 3): In your study, you assessed radiomics features associated with recurrence. I would like you to add the survival endpoints (OS, RFS, TTR, etc) in the group of patients who had recurrence and in those who did not have recurrence.
Point 4): Please add the values of survival times (OS, RFS, TTR, etc) in the IRE+resection vs IRE alone. This is important to understand the potential role of IRE on survival
Point 5): I agree that PMID 36837560 refers to neuroendocrine tumors, but you could add the concept. You could write "The application of radiomics to other alternative, less invasive local ablation methods (i.e., EUS-RFA) should be assessed in the future, also for different diseases than PDAC, such as neuroendocrine tumors".
Author Response
Point 3): In your study, you assessed radiomics features associated with recurrence. I would like you to add the survival endpoints (OS, RFS, TTR, etc) in the group of patients who had recurrence and in those who did not have recurrence. *** Thank, the overall survival for the IRE group has been added to the results section. If a patient has not had recurrence then they would not have a RFS (thus we have not presented these as two groups just a overall survival of RFS. The Time to response (TTR) would not apply for this study presented. ***
Point 4): Please add the values of survival times (OS, RFS, TTR, etc) in the IRE+resection vs IRE alone. This is important to understand the potential role of IRE on survival *** Again as above these have been added to the results. ***
Point 5): I agree that PMID 36837560 refers to neuroendocrine tumors, but you could add the concept. You could write "The application of radiomics to other alternative, less invasive local ablation methods (i.e., EUS-RFA) should be assessed in the future, also for different diseases than PDAC, such as neuroendocrine tumors". *** Thank you, but again this is a different cell type with completely different treatment modality, thus we do not want to confuse the reader or the results. We do not believe this additional reference supports our results and aim of the study. ***
Reviewer 2 Report
Comments and Suggestions for Authors
The authors have made the improvements that are required. From my point of view it can be published.
Author Response
Thank you for your review
Reviewer 3 Report
Comments and Suggestions for Authors
Thank you for making these changes, it has improved the clarity and strengthened the discussion. I have no further comments at this time.
Author Response
Thank you for your time to review
Round 3
Reviewer 1 Report
Comments and Suggestions for Authors
I have no further comments